# Recent Developments in Combination Chemotherapy for Colorectal and Breast Cancers with Topoisomerase Inhibitors

**DOI:** 10.3390/ijms24098457

**Published:** 2023-05-08

**Authors:** Jung Yoon Jang, Donghwan Kim, Nam Deuk Kim

**Affiliations:** 1Department of Pharmacy, College of Pharmacy, Research Institute for Drug Development, Pusan National University, Busan 46241, Republic of Korea; jungyoon486@pusan.ac.kr; 2Functional Food Materials Research Group, Korea Food Research Institute, Wanju-gun 55365, Republic of Korea; kimd@kfri.re.kr

**Keywords:** topoisomerases, irinotecan, doxorubicin, clinical trial, combination chemotherapy regimens

## Abstract

DNA topoisomerases are important enzymes that stabilize DNA supercoiling and resolve entanglements. There are two main types of topoisomerases in all cells: type I, which causes single-stranded DNA breaks, and type II, which cuts double-stranded DNA. Topoisomerase activity is particularly increased in rapidly dividing cells, such as cancer cells. Topoisomerase inhibitors have been an effective chemotherapeutic option for the treatment of several cancers. In addition, combination cancer therapy with topoisomerase inhibitors may increase therapeutic efficacy and decrease resistance or side effects. Topoisomerase inhibitors are currently being used worldwide, including in the United States, and clinical trials on the combination of topoisomerase inhibitors with other drugs are currently underway. The primary objective of this review was to comprehensively analyze the current clinical landscape concerning the combined application of irinotecan, an extensively investigated type I topoisomerase inhibitor for colorectal cancer, and doxorubicin, an extensively researched type II topoisomerase inhibitor for breast cancer, while presenting a novel approach for cancer therapy.

## 1. Introduction

Cancer is expected to become the leading cause of death and the most significant barrier to increasing life expectancy worldwide in the 21st century [1]. Cancer is an important public health issue worldwide and ranks second among the causes of death in the United States. In 2023, 1,958,310 new cancer cases and 609,820 cancer-related deaths are expected in the United States [2]. Therefore, the development of novel and more specific chemotherapeutic agents against the most aggressive tumors and the identification of new biological targets are vital goals in cancer research [1,3]. One of the main drug targets used in chemotherapy to inhibit the abnormal proliferation of cancer cells is topoisomerase (TOPO) [4].

DNA TOPOs are a group of enzymes that regulate DNA topology. TOPO activity increases especially in rapidly dividing cancer cells [5]. They are involved in many important cellular biological processes, including DNA replication, transcription, recombination, and chromosome condensation [6]. These enzymes covalently attach to groups of DNA phosphorus, causing the DNA strands to split and finally recombine [5]. Depending on the number of DNA strands cut, TOPO can be classified into types I and II. Type I enzymes cleave only one DNA strand, whereas type II enzymes cleave both strands to prevent supercoiling or entanglement [7]. Many anticancer drugs act as TOPO poison inhibitors that trap covalent complexes of human TOPOs, causing DNA damage and cancer cell death [8,9]. Clinically approved TOPO-targeting drugs include the camptothecin analog irinotecan (a prodrug of SN-38), topotecan, and belotecan as TOPO I inhibitors and pixantrone, etoposide, etopophos (etopoposide phosphate), teniposide, doxorubicin, epirubicin, valrubucin, daunorubucin, idarubicin, amrubicin, aclarubicin, amsacrine, and mitoxantrone as TOPO II inhibitors [5,10,11,12]. Despite their clinical efficacy, current anticancer therapies that use TOPO-directed agents have several important limitations and adverse effects. Traditional TOPO I inhibitors (e.g., such as camptothecin) exhibit significant dose-limiting toxicity and cancer cells develop resistance to these drugs. In addition, treatment with drugs targeting TOPO II can induce secondary malignancies such as acute myeloid leukemia due to its inhibition [13,14,15]. Therefore, new approaches are required to improve the efficacy of cancer treatment with TOPO inhibitors and counteract their side effects. One approach is combination therapy using a topoisomerase-targeting drug and another drug.

Combination therapy—the combination of two or more treatments—is the cornerstone of cancer treatment. Combinations of anticancer agents show improved efficacy compared to monotherapy because they characteristically target key pathways in a synergistic or additive manner. This approach potentially diminishes drug resistance while providing therapeutic anticancer benefits, such as reduced tumor growth, mitosis arrest, reduced metastatic potential, reduced cancer stem cell populations, and induction of apoptosis [16]. This review presents the clinical outcomes of combination therapies for colorectal cancer (CRC) using irinotecan, the most widely studied TOPO I inhibitor, and doxorubicin, the most widely studied TOPO II inhibitor, for breast cancer.

## 2. Combination Therapy

Combination therapies increase the efficacy of cancer treatments and cope with multiple genetic changes in different cancer cells. This involves administering one or more types of treatment simultaneously, such as two or more chemotherapies or a combination of chemotherapy and radiation/adjuvant therapy. Occasionally, one or more natural products with antitumor activities, such as low-molecular-weight components of herbs or fungi, may be used in combination therapies [17]. In addition, combination therapies can be applied in cancer cell cultures, animal xenograft models, and clinical trials of patients with cancer. Although the monotherapy approach remains an extremely common treatment for many types of cancer, this conventional method is usually considered less effective than the combined therapy approach [16]. Conventional single-treatment approaches target proliferating cells non-selectively, eventually destroying both healthy and cancer cells. The combination of two drugs may exhibit synergistic, antagonistic, or additive effects compared with their attributes in monotherapy. This approach is intended to minimize the side effects of monotherapy, such as chemical drug resistance, low efficacy, final dose reduction with biological effects, and other side effects leading to patient death [18].

## 3. TOPO I Inhibitors

### 3.1. TOPO I Mechanism and Inhibitors

TOPO I induces temporary breaks in a single strand of DNA, leading to changes in its topology. TOPO I can be divided into three subfamilies: type IA, IB, and IC [8,11,19]. Altering the DNA topology by breaking the phosphodiester bonds between nucleotides in DNA strands is based on the same general mechanism in all subtypes of TOPO I. The phosphoryl group of the DNA is attacked by the tyrosyl group of TOPO I, resulting in the formation of a covalent bond between the tyrosyl group and one side of the broken DNA. Simultaneously, the free hydroxylated strands unwind and rotate. The hydroxyl ends of the free DNA strand attack the produced phosphotyrosine bonds. The phosphodiester bonds between the two strands are reconstructed, and topoisomerase is released to participate in the next catalytic cycle [20]. In most cases, TOPO changes the phase of DNA by type 1, which does not require external energy (e.g., ATP hydrolysis) [21]. Type IA TOPOs require a nick- or a single-stranded region to bind to the DNA. They change the DNA topology by cleaving one strand of double-stranded DNA, covalently attaching an active site tyrosine to a 5′-phosphoryl group and utilizing a ‘strand passage’ mechanism. In contrast, type IB and IC TOPOs cleave one strand of double-stranded DNA, attach with the active site tyrosine to the 3′-phosphoryl group to form covalent bonds, and utilize a ‘controlled rotation’ mechanism to relax the DNA supercoil [8,22,23] (Figure 1A,B).

For a long time, camptothecins were the only class of compounds demonstrated to target TOPO I. Camptothecin (CPT) was isolated from the stem and bark of *Camptotheca acuminate* in 1966 by M. E. Wall and M. C. Wani in a natural product screening for anticancer drugs [24]. CPT interacts with DNA and TOPO I enzymes via a hydrogen bond to form the TOPO I–DNA–camptothecin ternary complex. This ternary complex collides with the DNA replication fork causing DNA damage and eventually leading to cell death [19]. Clinically approved TOPO I inhibitors include the camptothecin analog irinotecan (a prodrug of SN-38), topotecan, and belotecan [5,10,11,12].

### 3.2. Irinotecan Combination Therapy

Irinotecan is an extensively studied TOPO I inhibitor. The approval for the use of irinotecan (Camptosar^®^) as a treatment for cervical, lung, and ovarian cancer was granted in Japan in 1994, and in 1995 and 1996, it was approved for use in Europe and the United States, respectively [25] (Figure 1C). Irinotecan is a prodrug that is metabolically active in the body, similar to 7-ethyl-10-hydroxycamptothecin (SN-38) [26]. Irinotecan is used in the treatment of advanced CRC and other solid tumors, including pancreatic and non-small cell lung cancer, biliary tract cancer, and advanced gastric and cervical cancer. To date, various clinical trials have revealed the survival advantages of irinotecan-based therapy in patients with metastatic CRC, making it one of the main drugs used for the treatment of metastatic CRC [26]. It is used in pediatric and adult oncology. Although irinotecan can be used as monotherapy, it is used in combination with other cytotoxic agents, such as oxaliplatin and 5-fluorouracil, and monoclonal antibodies, such as bevacizumab and cetuximab. Experimental and clinical studies have shown that irinotecan can be combined with kinase inhibitors, such as apatinib, fruquintinib, dasatinib, regorafenib, and sunitinib, as well as cell cycle checkpoint inhibitors [27]. Irinotecan-based combinations vary widely. These drugs can be appropriately combined with DNA repair inhibitors, agents affecting epigenetic modifications, signal modulators, and immunotherapies [10].

### 3.3. Clinical Status of Irinotecan Combination Therapy in CRC

According to clinical trial reports, clinical studies on combination therapy with irinotecan are the most common in CRC. Therefore, we summarized the studies that reported the results of clinical trials of combination therapy with irinotecan in CRC (Table 1).

According to the 2023 United States cancer statistics, CRC is the third most commonly diagnosed cancer in both males and females and the third leading cause of estimated deaths in both sexes [2]. The treatment of CRC typically involves a combination of surgery, chemotherapy, and radiation therapy, depending on the stage and location of the cancer and the patient’s overall health and other individual factors [76]. Surgery is the primary treatment for CRC and involves the removal of the tumor and surrounding tissue. In some cases, the entire colon may require removal (colectomy). After surgery, chemotherapy can be administered to kill any remaining cancer cells and reduce the risk of cancer recurrence. The National Comprehensive Cancer Network (NCCN) guidelines recommend the use of chemotherapy regimens, including CAPOX (capecitabine and oxaliplatin), FOLFIRI (folinic acid, fluorouracil, and irinotecan), FOLFOX (folinic acid, fluorouracil, and oxaliplatin), or FOLFOXIRI (folinic acid, fluorouracil, oxaliplatin, and irinotecan), for unresectable metastatic CRC [77]. The most commonly used chemotherapeutic regimens for CRC are FOLFOX and FOLFIRI [78].

Many clinical studies on FOLFIRI and FOLFOXIRI combined with irinotecan for CRC treatment have been reported. The aim of the NCT01183780 trial, which has been referenced the most among clinical trials on FOLFIRI, was to evaluate the overall survival of metastatic CRC patients who received either ramucirumab plus FOLFIRI or placebo plus FOLFIRI [63]. Ramucirumab is a human IgG-1 monoclonal antibody that interacts with the extracellular part of the vascular endothelial growth factor (VEGF) receptor 2, which is important for blood vessel growth. Targeting angiogenesis is crucial for CRC treatment. Ramucirumab has been proven effective in treating several types of cancer, including gastric, lung, urothelial, colorectal, and advanced liver cancers [79]. The NCT01183780 study, which involved 1072 patients, showed that ramucirumab, in combination with FOLFIRI, as a second-line treatment for metastatic CRC, significantly enhanced the overall survival rate compared to placebo with FOLFIRI. Moreover, no unexpected negative events were observed, and the adverse effects were controllable [64]. In CRC, the identification of activating *RAS/RAF* mutations early in the disease is a crucial molecular discovery, and these mutations have been suggested as biomarkers for predicting treatment outcomes and disease prognosis [80]. In the NCT01183780 study, adding ramucirumab to FOLFIRI resulted in improved patient outcomes, regardless of *RAS/RAF* mutation status or tumor location [65,68].

## 4. TOPO II Inhibitors

### 4.1. TOPO II Mechanism and Inhibitors

TOPO II are enzymes that cleave both strands of the DNA double helix at the same time and are used to untangle and relieve supercoils in DNA [81]. There are two subtypes of TOPO II, TOPO IIA and TOPO IIB, which are found in different organisms. TOPO IIA exists in bacteria, eukaryotes, and a small number of archaea species, whereas TOPO IIB is mainly found in archaea, plants, and some algae [5]. TOPO IIA is primarily involved in DNA replication and mitosis, whereas TOPO IIB regulates gene expression during transcription. The activity of TOPO II (or TOPO IIA) during mitosis is crucial for the survival of cells [82]. The main mechanism through which TOPO II alters DNA topology involves cutting both DNA strands using Mg^2+^ and ATP hydrolysis. These enzymes can relax both positive and negative supercoils in DNA and pass a second DNA duplex through a gap after covalently attaching tyrosine to the 5′-end of broken DNA and releasing a free 3′-end (Figure 2A). TOPO II plays a vital role in various nuclear processes, including transcription, replication, and recombination, because of its exceptional ability to untangle double strands of DNA. Loss of TOPO II activity results in double-stranded DNA breaks and cell death, whereas increased DNA cleavage can lead to DNA translocation [5].

TOPO II inhibitors are categorized into two types based on their mode of action: catalytic inhibitors and TOPO II poisons. Catalytic inhibitors of TOPO II hinder its enzymatic functions. They obstruct the enzyme either before the cleavage of DNA or after the re-ligation of DNA is completed. As a result, these inhibitors do not cause the accumulation of TOPO II-DNA cleavage complexes. The lack of TOPO II activity in relaxing DNA supercoils or disentangling sister chromatids during mitosis can lead to unsuccessful cell division and, ultimately, cell death [83]. TOPO II poisons prevent TOPO II from completing the catalytic cycle after DNA cleavage. As a result, they increase the accumulation of TOPO II-DNA cleavage complexes, which can cause DNA damage that the cell’s DNA repair system cannot handle. This leads to the accumulation of DNA breaks, ultimately triggering programmed cell death. TOPO II poisons include etoposide, doxorubicin, and amsacrine [84].

### 4.2. Doxorubicin Combination Therapy

Doxorubicin is one of the most widely studied TOPO II inhibitors (Figure 2B). Doxorubicin was isolated from *Streptomyces peucetius* actinobacteria in the 1960s and was subsequently developed as a cancer drug [85]. Doxorubicin is a chemotherapeutic drug belonging to the anthracycline antibiotic family, with the trade name adriamycin. It is widely recognized as one of the most effective treatments for solid tumors and is used to treat several types of cancers, including breast cancer, bladder cancer, Kaposi’s sarcoma, lymphoma, and acute lymphocytic leukemia [86]. Doxorubicin is widely used for the treatment of breast cancer [87,88]. Doxorubicin was approved for medical use in the United States in 1974 [12]. It is considered an essential medicine by the World Health Organization [89]. Although doxorubicin is an effective chemotherapy for various types of malignant tumors, its application is limited owing to the risk of cardiotoxicity [90]. Therefore, doxorubicin is often used in combination with other drugs or therapies to increase its efficacy and reduce its side effects or the risk of drug resistance. One of the most commonly used doxorubicin-based combination therapies is the AC-T regimen, which involves a combination of doxorubicin and cyclophosphamide (AC), followed by taxane drugs, such as paclitaxel or docetaxel (T). This combination effectively reduces the risk of recurrence in early-stage breast cancer [91]. Doxorubicin can also be used in combination with targeted therapies, such as trastuzumab (Herceptin^®^) or pertuzumab (Perjeta^®^), in breast cancers that overexpress the human epidermal growth factor receptor 2 (HER2) protein. These targeted therapies function by blocking HER2 protein, which promotes the growth of cancer cells. When used in combination with doxorubicin, these targeted therapies can improve treatment effectiveness [92,93].

### 4.3. Clinical Status of Doxorubicin Combination Therapy in Breast Cancer

According to clinical trial reports, combination therapy with doxorubicin is the most widely studied treatment for breast cancer. Therefore, we have summarized the studies reporting the results of clinical trials on doxorubicin combination therapy for breast cancer (Table 2).

According to 2023 cancer statistics in the United States, breast cancer accounts for 31% of new diagnoses in women, ranking first, and is also the second leading cause of estimated deaths in women [2]. The primary objectives of treatment for breast cancer that has not spread to other parts of the body (non-metastatic) are to eliminate the tumor from the breast and nearby lymph nodes and prevent cancer from returning and spreading to other areas. Local treatment for nonmetastatic breast cancer typically involves surgery to remove the tumor and nearby lymph nodes; radiation therapy may also be considered after surgery [128]. The use of adjuvant chemotherapy is crucial in lowering the likelihood of breast cancer recurrence and enhancing the survival rate of patients. The NCCN’s guidelines for breast cancer treatment suggest several adjuvant chemotherapy plans, such as AC-T (sequential doxorubicin–cyclophosphamide and paclitaxel or docetaxel), ACT (concurrent doxorubicin–cyclophosphamide and paclitaxel or docetaxel), AC (doxorubicin–cyclophosphamide), CMF (cyclophosphamide, methotrexate, and fluorouracil), and TC (docetaxel and cyclophosphamide). Sequential AC-T therapy is the most widely used regimen [91].

Numerous clinical studies have reported the use of doxorubicin-based AC-T regimens for breast cancer treatment. The aim of the NCT00312208 study was to compare the disease-free survival of patients with operable breast cancer with positive axillary lymph nodes who were HER2-neu negative and treated either with docetaxel combined with doxorubicin and cyclophosphamide (TAC) or with doxorubicin and cyclophosphamide, followed by docetaxel (AC-T) [102]. The NCT00312208 study, which included 3299 patients, analyzed the data after 10 years and found that TAC was not more effective than AC-T in women with early-stage breast cancer and positive lymph nodes. The toxicity profiles of the two treatment groups were different, which is consistent with previous reports [103].

The aim of NCT00021255, which has the highest number of references among the studies of doxorubicin-based AC-T therapy, was to evaluate the disease-free survival of women diagnosed with operable breast cancer and showing HER2-neu expression with positive or high-risk node-negative lymph nodes. In this study, the researchers compared the effectiveness of two adjuvant treatment regimens during the treatment period: doxorubicin, cyclophosphamide, and docetaxel with or without trastuzumab, docetaxel, and carboplatin [108]. *HER2 (ERBB2)* is a member of the human type 1 receptor tyrosine kinases [109]. In a certain percentage of breast cancers (approximately 15–20%), this gene is amplified, leading to the overexpression of the HER2 protein, resulting in the transformation of normal cells to cancerous cells [129,130]. Normally, HER2 is activated only when a ligand binds to one of the other three members of the HER family—epidermal growth factor receptor (EGFR)/HER1, HER3, or HER4)—leading to the formation of heterodimers with HER2 and the activation of its kinase activity [131]. However, when HER2 is overexpressed, it associates with itself and other HER family members in a ligand-independent manner [109]. Trastuzumab is a monoclonal antibody used to treat breast cancer overexpressing HER2 [132]. The NCT00021255 study, which involved 3222 patients, showed that the addition of adjuvant trastuzumab for one year resulted in significant improvements in disease-free and overall survival rates among women diagnosed with HER2-positive breast cancer [113]. Additionally, the loss of the tumor suppressor gene phosphatase and tensin homolog (PTEN) is associated with a worse prognosis in patients with HER2-amplified breast cancer; however, this is not related to trastuzumab resistance. This study demonstrated that PTEN deficiency is not a predictive factor for trastuzumab resistance in HER2-positive breast cancer [109].

## 5. Conclusions

In this review, we described the clinical status of combination chemotherapy for CRC, primarily using irinotecan, the most extensively studied TOPO I inhibitor, and for breast cancer, primarily using doxorubicin, the most extensively studied TOPO II inhibitor. DNA replication, transcription, and repair are essential for every cell, and TOPOs play crucial roles in these processes. Owing to their significant biological functions, enzyme structures, and mechanisms of action, TOPOs have been a major focus in the development of novel anticancer agents. Combination chemotherapy with TOPO inhibitors induces cellular stress and cell death by causing cell cycle arrest, apoptosis, autophagy, and necroptosis pathways (Figure 3). However, TOPO inhibitors are subject to drug resistance, have significant dose-limiting toxicity, and can induce secondary cancers. Therefore, clinical studies on combination chemotherapy using TOPO inhibitors, together with other cancer therapeutic agents, are continually evolving to decrease these phenomena. Examples of current studies include ABVD (doxorubicin, bleomycin, vinblastine, and dacarbazine) for Hodgkin’s lymphoma, CBV (cyclophosphamide, carmustine, and etoposide) for lymphoma, and CAV (cyclophosphamide, doxorubicin, and vincristine) for small cell lung cancer. These studies demonstrate new potential for cancer treatment using TOPO inhibitors.

## Figures and Tables

**Figure 1 ijms-24-08457-f001:**
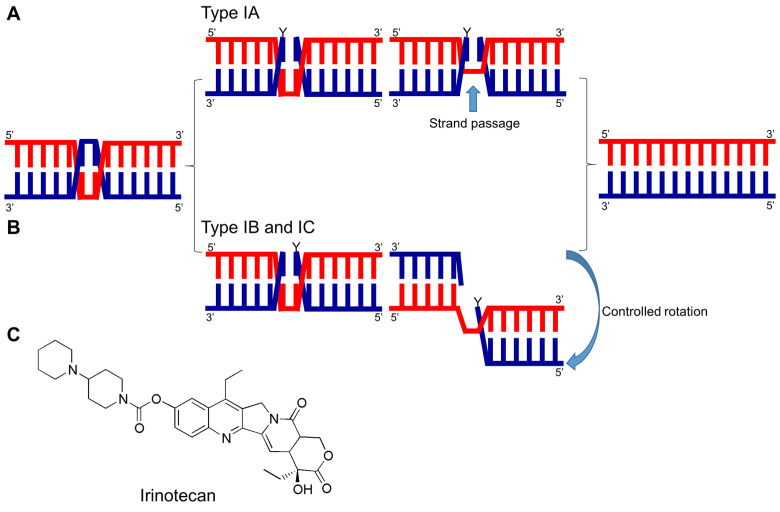
Catalytic mechanism of type I DNA TOPOs and chemical structure of irinotecan. (**A**) Each of the three types of DNA TOPOs has a distinct mechanism for catalyzing changes in DNA topology. Type IA functions as a monomer, cleaving one DNA strand and creating a 5′-phospho-tyrosyl bond within the protein-DNA complex. This creates an opening in the cleaved strand, allowing the uncut strand to pass through for relaxation or decatenation of the DNA. The ends of the cut strand are then reconnected, restoring the DNA backbone, and the enzyme can dissociate from the 5′-end of the DNA. (**B**) Type IB and IC also act as monomers but cleave one strand of duplex DNA and form a temporary 3-phospho-tyrosyl bond. DNA relaxation is achieved by the controlled rotation of the free 5′-end of the DNA around the uncut strand. (**C**) Chemical structural formula of irinotecan.

**Figure 2 ijms-24-08457-f002:**
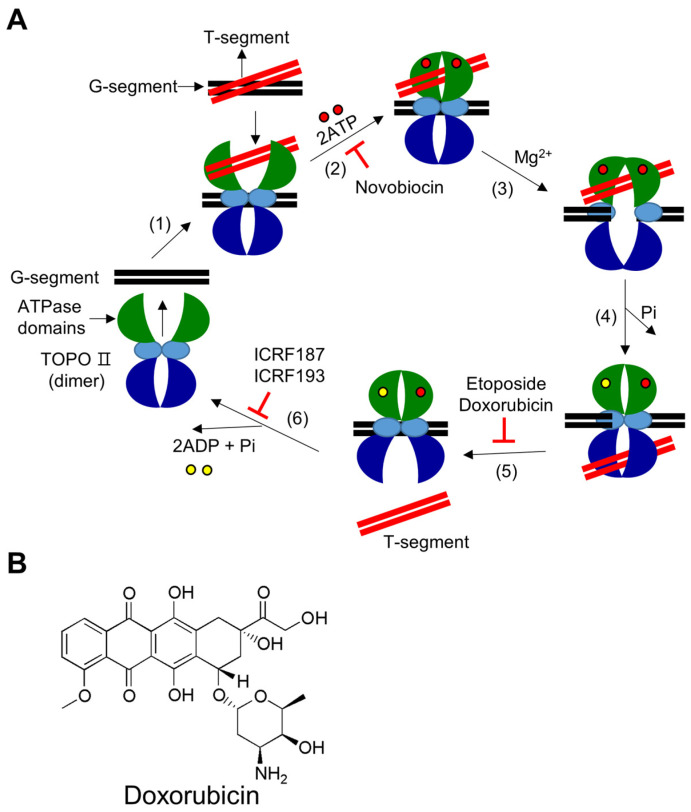
Catalytic mechanism of type II DNA TOPOs and chemical structure of doxorubicin. (**A**) (1) DNA binding: The enzyme’s homo-dimer preferentially binds to catenated, knotted, and supercoiled DNA segments. The segment of double-stranded DNA that is cleaved during the enzymatic reaction cycle is referred to as the “G segment” (with “G” for gate), and the segment of double-stranded DNA that passes through the cleaved G segment is referred to as the “T segment” (with “T” for transported). The enzyme binds to the G segment and then to the T segment. (2) ATP binding: The binding of two ATP molecules in the ATPase domains alters the conformation of the ATPase domains from an open to a closed state. Novobiocin prevents ATP binding. (3) DNA cleavage: In the presence of Mg^2+^ ions, the enzyme temporarily cleaves the G segment of DNA by initiating a nucleophilic attack and forming two 5′-phosphotyrosyl bonds with the DNA backbone. (4) Strand passage: After the G segment is cleaved, the T segment is threaded through it. (5) T segment release and re-ligation: Once the T segment has passed through, it is released from the enzyme, and the cleaved G segment is rejoined. Etoposide and doxorubicin prevent the rejoining process. (6) G segment releases when the ATPase domain is opened: After the T segment is released, the enzyme stays in a closed clamp shape. Hydrolysis of ATP causes the closed clamp to open, allowing the G segment to be released and preparing the enzyme for the next reaction cycle. Bisdioxopiperazines, such as ICRF187 and ICRF193, inhibit the ATPase activity of the enzyme. (**B**) Chemical structural formula of doxorubicin.

**Figure 3 ijms-24-08457-f003:**
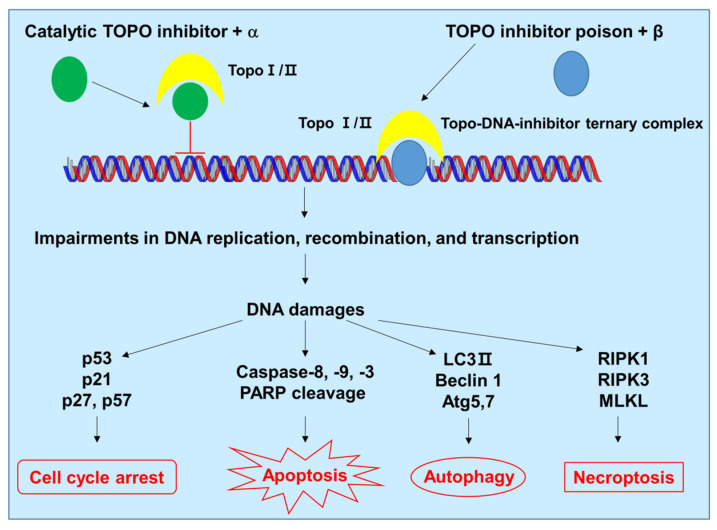
Proposed model of cell death induced by combination of chemotherapeutic drugs and TOPO inhibitors. The combination chemotherapy causes DNA damage, generates cellular stress, and induces cell death by activating cell cycle arrest, apoptosis, autophagy, and necroptosis pathways. α and β refer to other chemotherapeutic drugs used in combination therapy. Atg, autophagy related; LC3, microtubule-associated protein 1A/1B-light chain 3; MLKL, mixed lineage kinase domain-like protein; RIPK, receptor-interacting protein kinase; PARP, poly(ADP-ribose) polymerase.

**Table 1 ijms-24-08457-t001:** Clinical status of irinotecan combination therapy in CRC.

Drugs	Target Cancer	Purpose of Study	Clinical Trials Status	Clinical Trials Identifier	Refs.
Capecitabine, Irinotecan	CRC	A study on the combined effect of capecitabine and irinotecan	Phase 2	NCT00022698	[28,29,30]
Ascorbic acid, Irinotecan	Stage IV CRC	Phase I/II study of ascorbic acid infusion versus irinotecan monotherapy combined with irinotecan treatment in patients with relapsed or advanced CRC who have failed at least one treatment regimen with a fluorouracil-based regimen	Phase 1 Phase 2	NCT01550510	[31]
Pemetrexed, Irinotecan	Metastatic CRC	To determine the efficacy and safety of the combination of pemetrexed and irinotecan	Phase 2	NCT00191984	[32,33]
ISIS 183750, Irinotecan	Colorectal neoplasms Colorectal carcinoma Colorectal tumors	Testing the safety and efficacy of irinotecan against ISIS 183750 and advanced solid or CRC	Phase 1 Phase 2	NCT01675128	[34,35,36]
MM-121, Irinotecan, Cetuximab	CRC	To evaluate the safety and tolerability of escalating doses of MM-121 + cetuximab and MM-121 + cetuximab + irinotecan combinations	Phase 1	NCT01451632	[37,38]
S-1, Irinotecan, Bevacizumab	CRC	To determine if it is safe to treat unresectable or recurrent colorectal cancer	Phase 2	NCT00569790	[39,40]
Tivantinib, Cetuximab, Irinotecan	Metastatic CRC	ARQ 197 or placebo plus irinotecan and cetuximab, defines the recommended dose for phase 2	Phase 1 Phase 2	NCT01075048	[41,42]
Guadecitabine (SGI-110), Regorafenib, Lonsurf (TAS-102), Irinotecan	Previously treated metastatic CRC	Enrollment in phase 1 study of SGI-110 combined with irinotecan and after the MTD was determined, patients were enrolled in a 2:1 randomized phase 2 study of SGI-110 and irinotecan versus standard-of-care regorafenib or TAS-102	Phase 1 Phase 2	NCT01896856	[43,44]
Cetuximab, FOLFIRI	EGFR expressing metastatic CRC	To investigate the effect of cetuximab in combination with chemotherapy (FOLFIRI) compared to the same chemotherapy for patient EGF receptors	Phase 3	NCT00154102	[45,46,47,48,49]
Regorafenib, FOLFIRI	CRC metastatic	Comparison of PFS between regorafenib + FOLFIRI chemotherapy and placebo + FOLFIR in mCRC patients previously treated with FOLFOX therapy	Phase 2	NCT01298570	[50,51,52]
ABT-165, Bevacizumab, FOLFIRI	Previously treated metastatic adenocarcinoma of the colon or rectum	Study evaluating efficacy and tolerability of ABT-165 + FOLFIRI compared to bevacizumab + FOLFIRI	Phase 2	NCT03368859	[53,54]
Encorafenib, Binimetinib, Cetuximab, FOLFIRI	BRAF V600E-mutant metastatic CRC	To evaluate encorafenib plus cetuximab plus or minus binimetinib versus choosing either irinotecan/cetuximab or FOLFIRI/cetuximab as control in patients with BRAFV600E mCRC	Phase 3	NCT02928224	[55,56,57,58,59,60]
Napabucasin, Bevacizumab, FOLFIRI	CRC	Trial of cancer stem cell pathway inhibitor napabucacin plus standard biweekly FOLFIRI versus standard biweekly FOLFIRI	Phase 3	NCT02753127	[61]
FOLFIRI, Panitumumab	Recurrent colorectal carcinoma Stage IVA CRC Stage IVB CRC	A study on how well FOLFIRI works in combination with panitumumab in the treatment of CRC patients	Phase 2	NCT02508077	[62]
Ramucirumab, FOLFIRI	CRC	Comparing overall survival of participants with metastatic CRC treated with ramucirumab plus FOLFIRI or placebo plus FOLFIRI	Phase 3	NCT01183780	[63,64,65,66,67,68,69,70]
Bevacizumab, FOLFOXIRI	Colorectal neoplasms	Efficacy and safety evaluation of FOLFOXIRI/bevacizumab regimen (concurrent and sequential) versus FOLFOX/bevacizumab	Phase 2	NCT01765582	[71,72,73]
Panitumumab, FOLFOXIRI	CRC	A plan to determine the ORR of the combination of FOLFOXIRI and panitumumab	Phase 2	NCT01226719	[74,75]

EGF, epidermal growth factor; EGFR, epidermal growth factor receptor; FOLFIRI, folinic acid, fluorouracil, and irinotecan; FOLFOX, folinic acid, fluorouracil, and oxaliplatin; FOLFOXIRI, folinic acid, fluorouracil, oxaliplatin, and irinotecan; ORR, objective response rate; PFS, progression-free survival.

**Table 2 ijms-24-08457-t002:** Clinical status of doxorubicin combination therapy in breast cancer.

Drugs	Target Cancer	Purpose of Study	Clinical Trials Status	Clinical Trials Identifier	Refs.
Cisplatin, AC	Breast cancer	To evaluate cisplatin, a chemotherapy drug that has been shown to be active in the treatment of breast cancer and women with BRCA mutations	Phase 2	NCT01670500	[94,95,96,97]
Eribulin, AC	Inflammatory breast cancer HER2-negative carcinoma of breast	Studying a drug called eribulin combined with standard therapy as a possible preoperative treatment for HER2-negative inflammatory breast cancer	Phase 2	NCT02623972	[98]
Bevacizumab, Paclitaxel, Gemcitabine hydrochloride, Pegfilgrastim, AC	HER2-negative breast cancer Stage II breast cancer Stage IIIA breast cancer Stage IIIB breast cancer Stage IIIC breast cancer	To investigate the efficacy and side effects of adding bevacizumab to the chemotherapy regimen in the treatment of stage 2 or 3 HER2-neu negative breast cancer in women	Phase 2	NCT00679029	[99]
AC, Paclitaxel, Tipifarnib	Breast cancer Male breast cancer	To study the side effects and optimal dose of tipifarnib when given with combination chemotherapy and how effective it is in treating patients with stage 2 or 3 breast cancer	Phase 1 Phase 2	NCT00470301	[100,101]
AC-T (Docetaxel)	Breast cancer	Comparison of disease-free survival after TAC versus AC-T in HER2-neu negative breast cancer patients who are eligible for surgery	Phase 3	NCT00312208	[102,103,104]
AC-T (Paclitaxel), Ixabepilone	Breast cancer	Comparison of patients receiving AC and ixabepilone and patients receiving AC and weekly paclitaxel	Phase 3	NCT00789581	[105,106]
AC-T (Paclitaxel), Epoetin alfa, Filgrastim, Epirubicin hydrochloride, Fluorouracil	Breast cancer	Comparing the effectiveness of chemotherapy with or without epoetin alfa for the treatment of women who have undergone surgery for stage I, II, or III breast cancer	Phase 3	NCT00014222	[107]
AC-T (Docetaxel), Trastuzumab, Carboplatin	Breast neoplasms	Comparing the treatment outcomes of women with HER2-positive breast cancer who had positive lymph nodes or high-risk negative lymph nodes and were treated with adjuvant therapy including doxorubicin, cyclophosphamide, and docetaxel, with or without trastuzumab, versus those who received trastuzumab, docetaxel, and carboplatin	Phase 3	NCT00021255	[104,108,109,110,111,112,113]
FAC, Docetaxel	Breast cancer	Comparison of disease-free survival rates after TAC combination therapy and FAC combination therapy	Phase 3	NCT00688740	[114,115,116]
FAC, Paclitaxel, Eribulin, Epirubicin	Breast cancer	To find out if and how well eribulin given in combination with standard chemotherapy can treat early-stage breast cancer compared to paclitaxel given in combination with standard chemotherapy	Phase 2	NCT01593020	[117,118]
Doxorubicin, FEC	Breast cancer	Two combination chemotherapy regimens were studied to compare how effective they were in treating women who had surgery for breast cancer that had not spread to the lymph nodes	Phase 3	NCT00087178	[119]
TAC	Breast cancer	To see if we can find out if taxotere and/or adriamycin/cytoxan can make tumors smaller	Phase 2	NCT00206518	[120]
TAC	Breast cancer	To find out what effect (good or bad) TC or TAC has on early-stage HER2- breast cancer	Phase 3	NCT00493870	[121,122]
TAC, Paclitaxel, Gemcitabine	Breast cancer	Studying three different combination chemotherapy regimens and comparing how effective they are in treating women who have had surgery for node-positive breast cancer	Phase 3	NCT00093795	[123,124]
AC-TH, Carboplatin	Breast cancer	To evaluate the safety of trastuzumab for the treatment of HER2-positive nodule-positive or high-risk nodule-negative	Phase 4	NCT02419742	[125]
AC-THP, Atezolizumab, Trastuzumab emtansine	Breast cancer	To evaluate the efficacy and safety of atezolizumab compared with placebo when given in combination with neoadjuvant dose-dense doxorubicin + cyclophosphamide followed by paclitaxel + trastuzumab + pertuzumab in patients with early HER2-positive breast cancer	Phase 3	NCT03726879	[126,127]

AC, doxorubicin and cyclophosphamide; AC-T, sequential doxorubicin–cyclophosphamide and paclitaxel or docetaxel; AC-TH, doxorubicin plus cyclophosphamide, followed by paclitaxel plus trastuzumab; AC-THP, doxorubicin and cyclophosphamide, followed by paclitaxel, trastuzumab, and pertuzumab; FAC, fluorouracil, doxorubicin and cyclophosphamide; FEC, fluorouracil, epirubicin and cyclophosphamide; HER2, human epidermal growth factor receptor 2; TAC, docetaxel, doxorubicin, and cyclophosphamide; TC, docetaxel and cyclophosphamide.

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
