# Peer review of "Recent Developments in Combination Chemotherapy for Colorectal and Breast Cancers with Topoisomerase Inhibitors"

_ijms, 2023, doi:10.3390/ijms24098457_

Round 1
Reviewer 1 Report
This review summarized the recombinant treatment of topoisomerase inhibitors with other chemotherapeutic drugs in the treatment of colorectal and breast cancers in different clinical phases. The background of topoisomerase and inhibitors are thoroughly described. I suggest authors put a summary graph or two (TOPOI and TOPOII) in the conclusion section to illustrate which pathway or proteins were targeted or inhibited based on the papers which listed in this review. In this way, the paper will give readers perspective on how the combination treatment work and what can be further studied in the future.
Author Response
Response to Reviewer 1 Comments
Dear editor Ms. Codruta Cormos and reviewers:
Thank you for your letter and for the reviewers’ comments concerning our manuscript entitled “Recent Developments in Combination Chemotherapy for Colorectal and Breast Cancers with Topoisomerase Inhibitors” (ID: ijms-2340083). These comments are all valuable and very helpful for revising and improving our paper, as well as the important guiding significance to our researches. We have studied comments carefully and have made corrections which we hope meet with approval. Revised portions are marked in red on the paper. The main corrections in the paper and the response to the reviewer’s comments are as flowing:
ï¾· Point 1: This review summarized the recombinant treatment of topoisomerase inhibitors with other chemotherapeutic drugs in the treatment of colorectal and breast cancers in different clinical phases. The background of topoisomerase and inhibitors are thoroughly described. I suggest authors put a summary graph or two (TOPOI and TOPOII) in the conclusion section to illustrate which pathway or proteins were targeted or inhibited based on the papers which listed in this review. In this way, the paper will give readers perspective on how the combination treatment work and what can be further studied in the future.
ï¾· Response 1: Thanks for your comments. We have added Figure 3 to the conclusions section of the manuscript, illustrating that the combination chemotherapy with TOPO inhibitors targets specific pathways and proteins.
We tried our best to improve the manuscript and made some changes in the manuscript. These changes will not influence the content and framework of the paper. And here we did not list the changes but marked them in red in the revised paper.
We appreciate for Editors/Reviewers’ warm work earnestly and hope that the correction will meet with approval.
Once again, thank you very much for your comments and suggestions.

Reviewer 2 Report
Provide references for line 73.
Summarize information for each study of Table 1 and Table 2. Include informative information in the tables (e.g. as provided for NCT01183780 (lines 170-178), also NCT00312208 and NCT00021255. The additional information would improve greatly the usefulness of the review.
The summaries should include quantitative information (e.g. ... significant improvements (what is significant?) in. disease-free and overall survival rates ....).
Were relationships with patients' age and gender uncovered?
What is the anticipated future trend in the field, concerning topoisomerase inhibitors, for various different cancers relative to breast cancers and colorectal cancers?
Author Response
Response to Reviewer 2 Comments
Dear editor Ms. Codruta Cormos and reviewers:
Thank you for your letter and for the reviewers’ comments concerning our manuscript entitled “Recent Developments in Combination Chemotherapy for Colorectal and Breast Cancers with Topoisomerase Inhibitors” (ID: ijms-2340083). These comments are all valuable and very helpful for revising and improving our paper, as well as the important guiding significance to our researches. We have studied comments carefully and have made corrections which we hope meet with approval. Revised portions are marked in red on the paper. The main corrections in the paper and the response to the reviewer’s comments are as flowing:
ï¾· Point 1: Provide references for line 73.
Response 1: Thank you for your advice. The content of reference 17 is included in Lines 68-75. Therefore, we have rearranged the citation positions in the manuscript according to your suggestion.
ï¾· Point 2: Summarize information for each study of Table 1 and Table 2. Include informative information in the tables (e.g. as provided for NCT01183780 (lines 170-178), also NCT00312208 and NCT00021255. The additional information would improve greatly the usefulness of the review.
Response 2: Thank you for your comments. While it would be great to provide additional explanations for all the clinical trials mentioned in the table, it would make the content too lengthy. Therefore, we provided additional information on the FOLFIRI regimen, which has undergone many clinical studies for irinotecan-based combination therapy, and specifically on NCT01183780, which has the highest number of references. We added the background for why we chose to explain NCT01183780 in our manuscript.
Furthermore, since the AC-T regimen has been extensively studied in breast cancer, we explained NCT00312208 and NCT00021255. We also provided additional explanations for NCT00312208 in order to compare the difference between the AC-T regimen and the TAC regimen (Line 284~ Line 288). And we also provided additional explanations for NCT00021255, which has the highest number of references among the AC-T regimens. We added the background for why we chose to explain NCT00021255 in our manuscript.
ï¾· Point 3: The summaries should include quantitative information (e.g. ... significant improvements (what is significant?) in. disease-free and overall survival rates ....).
Response 3: Thank you for your suggestion. While it would be ideal to include quantitative information in the summary, it is too much to include in the table. Therefore, we have included the NCT numbers and references.
ï¾· Point 4: Were relationships with patients' age and gender uncovered?
Response 4: Thank you for your comments. As a result of our research, it seems that the relationship between the age and gender of the patient has not yet been revealed. We will see if additional studies of this relationship are reported in the future.
ï¾· Point 5: What is the anticipated future trend in the field, concerning topoisomerase inhibitors, for various different cancers relative to breast cancers and colorectal cancers?
Response 5: Thank you for your comments. Topoisomease inhibitors are currently used in combination therapy in breast cancer and colorectal cancers as well as other cancers.
1) Hodgkin's lymphoma: ABVD (doxorubicin, bleomycin, vinblastine, and dacarbazine)
2) Lmphoma: CBV (cyclophosphamide, carmustine, and etoposide)
3) Lung cancer: CAV (cyclophosphamide, doxorubicin, and vincristine)
Therefore, it is anticipated that topoisomerase inhibitors will continue to be widely used in the treatment of various types of cancer. We have added these details to the conclusion of the manuscript.
We tried our best to improve the manuscript and made some changes in the manuscript. These changes will not influence the content and framework of the paper. And here we did not list the changes but marked them in red in the revised paper.
We appreciate for Editors/Reviewers’ warm work earnestly and hope that the correction will meet with approval.
Once again, thank you very much for your comments and suggestions.

Round 2
Reviewer 2 Report
Author's revision responses were modest.